# PARTIAL TRACE-CLASS BAYESIAN NEURAL NETWORKS

## ABSTRACT

Bayesian neural networks (BNNs) allow rigorous uncertainty quantification in deep learning, but often come at a prohibitive computational cost. We propose three different innovative architectures of partial trace-class Bayesian neural networks (PaTraC BNNs) that enable uncertainty quantification comparable to standard BNNs but use significantly fewer Bayesian parameters. These PaTraC BNNs have computational and statistical advantages over standard Bayesian neural networks in terms of speed and memory requirements. Our proposed methodology therefore facilitates reliable, robust, and scalable uncertainty quantification in neural networks. The three architectures build on trace-class neural network priors which induce an ordering of the neural network parameters, and are thus a natural choice in our framework. In a numerical simulation study, we verify the claimed benefits, and further illustrate the performance of our proposed methodology on a real-world dataset.

## 1 INTRODUCTION

Neural networks are important and powerful tools for the prediction and modelling of complex data. Often, one large concern is in quantifying the uncertainty in the predictions of such a network. Bayesian neural networks (BNNs) can be used to quantify this uncertainty (Neal, 1996; Izmailov et al., 2021; Fortuin et al., 2021; Fortuin, 2022; Papamarkou et al., 2024), however the uncertainty quantification often comes with a high computational cost. In an attempt to reduce the sometimes prohibitively expensive running of BNNs, we wish to exploit the computational speed and efficiency of training and running a standard neural network, and combine this with a Bayesian model in a way that facilitates meaningful uncertainty quantification. With a similar goal in mind of capturing uncertainty at a lower cost, previously, the idea of only applying Bayesian inference on the last layer of a network has been studied in both the context of Bayesian Optimization Snoek et al. (2015) as well as in our context of Bayesian Neural Networks, Zeng et al. (2018) and Brosse et al. (2020), whose results suggested there is not much benefit to having any more than a single uncertainty layer within the model. In contrast to these *hybrid* approaches, which allow layers to be only either fully Bayesian or completely non-Bayesian (Valentin Jospin et al., 2020; Chang, 2021; Prabhudesai et al., 2023), a *partial Bayesian Neural Network* (pBNN) approach (Calvo-Ordoñez et al., 2024), also known as subnetwork inference (Izmailov et al., 2020; Daxberger et al., 2021b), allows layers to be split into Bayesian and non-Bayesian parameters. Andrade & Sato (2024) investigate different strategies when selecting Bayesian parameters in the neural network, and show that pBNNs can even improve the predictive power of BNNs.

Unlike standard BNNs, trace-class Bayesian neural networks (Sell & Singh, 2023) introduce a natural ordering of the prior weights and thus naturally lend themselves to truncation. We will recall the definition and some properties of trace-class BNNs in Section 2.1, before investigating *partial trace-class Bayesian neural networks* (PaTraC BNNs), which we introduce in Section 3. In particular, we look at three different potential structures for such PaTraC BNNs, and evaluate their effectiveness in a numerical study in Section 4, as well as on the abalone data set (Nash et al., 1994) in Section 5. A particular focus of this study lies on the PaTraC BNN's ability to accurately quantify the uncertainty in the target function of interest.

Our approach differs from these existing pBNNs in two important ways. First, we employ a trace-class BNN prior (Sell & Singh, 2023) which by construction enforces a natural ordering of the

network parameters, and which further suggests a node-based procedure to select Bayesian parameters. Second, we select *nodes* based on their relative importance in the trained network, and turn the associated weights into Bayesian parameters; rather than selecting parameters on their respective values after training or picking full layers to be Bayesian. An exception to this is our out-PaTraC BNN, which is more similar to the construction used by Franssen & Szabó (2022), where we focus only on a subset of the final layer weights to be Bayesian.

Other relevant work related to ours include variational inference for BNNs (Graves, 2011; Wu et al., 2018), which can be considered the state-of-the-art in BNN inference due to its low computational cost, although it comes at a relatively high price in terms of posterior approximation (Foong et al., 2020). Laplace approximations to BNNs (Daxberger et al., 2021a) also reduce the computational cost of full BNN inference. Different priors for BNNs are discussed e. g. by Wenzel et al. (2020); Noci et al. (2021); Fortuin (2022).

## 2 STATISTICAL SETTING

Given an input dimension $d \in \mathbb{N}$ and output dimension $m \in \mathbb{N}$, we are interested in estimating a function $f : \mathbb{R}^d \to \mathbb{R}^m$. Suppose there exists a true function $f^\star$ and a data generating mechanism which gives rise to the likelihood function $\mathcal{L}(f|\mathfrak{D})$, where we denoted $\mathfrak{D}$ the observed data set. For estimation purposes, we can now define a function class $\mathcal{F}$, from which the classical maximum likelihood estimation approach is to find a function $\hat{f}_{\mathrm{MLE}} \in \mathcal{F}$ which maximises the likelihood, i. e. $\hat{f}_{\mathrm{MLE}} := \mathrm{argmax}_{f \in \mathcal{F}} \mathcal{L}(f|\mathfrak{D})$. Due to the monotonicity of the logarithm, this is, of course, equivalent to minimising the negative log-likelihood, which we call the *loss function* $L(f) := -\log(\mathcal{L}(f|\mathfrak{D}))$.

The choice of the function class $\mathcal{F}$ is of key importance to the practitioner, this involves considerations such as model interpretability, perceived complexity of the target function to be estimated, computational cost of available inference methods, and flexibility of the model. For the purposes of this work, we are working with a nonparametric function class $\mathcal{F}$ imposed by a choice of a neural network structure. Such function classes have been shown to be capable of estimating a wide range of functions to an arbitrary level of precision and are now ubiquitously used across disciplines (Härdle, 1990). One further major advantage of using neural networks is that inference can be easily done using many existing optimisation libraries.

One of the limitations of optimising the neural network weights to estimate the function of interest is that it does not provide understanding of the uncertainty around the estimate, i. e. it could be that another neural network would be similarly capable at estimating the true function of interest. Furthermore, trained neural networks are known to be overconfident even in regions with lower amounts of training data (Kristiadi et al., 2020), and a desirable property of an estimation procedure would be honest about the uncertainty around the estimated function, which should generally be larger in regions with little training data. One principled approach to tackle these issues is given by the Bayesian paradigm, which yields an entire distribution over possible neural networks, giving more *posterior probability* to networks which fit the data well and are a priori probable.

More precisely, a *prior distribution $P(f)$* over all neural networks is defined by placing prior distributions on the weights and biases of the network, which are then weighted by the likelihood to give rise to the *posterior distribution*, which is well defined for a large range of priors even if taking the number of network nodes to infinity (Neal, 1996; de G. Matthews et al., 2018). In what follows, we define the prior distribution used throughout this paper, and then explain how to infer the posterior distribution $P(f|\mathfrak{D}) \propto \mathcal{L}(f|\mathfrak{D})P(f)$ using Markov chain Monte Carlo methods.

### 2.1 TRACE-CLASS BAYESIAN NEURAL NETWORK PRIOR

We use the trace-class neural network prior (Sell & Singh, 2023), which, like other Bayesian neural network priors, is well-defined in the infinite-width limit, but retains weights bounded away from 0 and thus allows one to interpret the (non-zero) function on the last network layer as hidden features – a desirable property for interpretation purposes (Neal, 1996). Additionally, posterior inference can be made more efficient as described in Section 2.2, and the asymmetry in the prior reduces multimodality in the posterior distribution of the network (Sell & Singh, 2023). Let $L$ be the total

number of layers in the network including the output layer, and $N^{(l)}$ the number of nodes in the $l$th layer, for $l = 1, \ldots, L$, additionally we define $N^{(0)} := d$. We write the pre-activated function on the $i$th node in the $l$th network layer as

$$f_i^{(l)}(x) = b_i^{(l)} + \sum_{j=1}^{N^{(l-1)}} w_{i,j}^{(l)} \zeta\big(f_j^{(l-1)}(x)\big),$$

where we denote the bias of the $i$th node of layer $l$ as $b_i^{(l)}$, the weight for the $j$th output of the layer $l - 1$ to the $i$th node of layer $l$ as $w_{i,j}^{(l)}$, and where $\zeta$ is the activation function. We summarise all neural network parameters as $\theta = (w_{i,j}^{(l)}, b_i^{(l)})_{i=1,j=1,l=1}^{N^{(l)},N^{(l-1)},L}$. While we choose the same activation for all layers, one may generally choose layer-dependent activation functions $\zeta^{(l)}$. The trace-class prior for the weights and biases is now defined by letting

$$w_{i,j}^{(1)} \sim N\left(0, \frac{\sigma_{w^{(1)}}^2}{i^\alpha}\right), \qquad w_{i,j}^{(l)} \sim N\left(0, \frac{\sigma_{w^{(l)}}^2}{(ij)^\alpha}\right) \quad \text{for } l \geq 2, \qquad b_i^{(l)} \sim N\left(0, \frac{\sigma_{b^{(l)}}^2}{i^\alpha}\right),$$

where $\alpha > 1$, $\sigma_{w^{(l)}}^2 > 0$ and $\sigma_{b^{(l)}}^2 > 0$ are parameters to be specified. This forces the nodes in the network which are 'further down' in a layer, i.e. those with larger indices, to have smaller variances. The exception to this is the first layer, where the inputs should be given equal prior importance and thus the variances for $w_{i,j}^{(1)}$ do not depend on $j$. The intuition behind the decreasing variances is that we *a priori* believe some weights to be larger and others to be closer to 0, and the proposed prior enforces this resulting in non-interchangeable nodes, with those nodes at the beginning of a layer (i.e. small $i$ and $j$) carrying more information than those further down in the layer (larger $i, j$). This remains true when increasing the number of nodes in a layer, and even in the infinite-width limit retains interpretable hidden features. We summarise all the variances in the covariance operator $\mathcal{C}$.

## 2.2 POSTERIOR INFERENCE

As the posterior distribution over the neural network parameters is analytically intractable, we resort to a Markov chain Monte Carlo (MCMC) method to approximate the posterior distribution. Even in the infinite width limit, the weights and biases can be shown to fall into the Hilbert space $\ell^2 = \{(a_1, a_2, \ldots) \in \mathbb{R}^{\mathbb{N}} : \sum_{i=1}^{\infty} a_i^2 < \infty\}$ (Sell & Singh, 2023, Lemma 5), such that we choose a dedicated Hilbert space MCMC method for posterior inference – the preconditioned Crank-Nicholson Langevin (pCNL) algorithm (Cotter et al., 2013). Given a current set of weights and biases $u$, the method proposes a move to $v$ by

$$v = \frac{1}{2 + \delta}\left((2 - \delta)u + 2\delta \mathcal{C}\mathcal{D}\ell(u) + \sqrt{8\delta}w\right),$$

where $w \sim N(0, \mathcal{C})$ is a random Gaussian vector, $\delta \in (0, 2)$ is a tuning parameter, $\mathcal{C}$ is the prior covariance operator defined in Section 2.1, and $\ell$ is the log-likelihood. The acceptance probability is given by $\min\{1, \exp(\rho(u, v) - \rho(v, u))\}$, where

$$\rho(u, v) = -\ell(u) - \frac{1}{2}\langle u - v, \mathcal{D}\ell(u)\rangle - \frac{\delta}{4}\langle v + u, \mathcal{D}\ell(u)\rangle + \frac{\delta}{4}\|\sqrt{\mathcal{C}}\mathcal{D}\ell(u)\|^2.$$

In our neural network setting, we have $\mathcal{D}\ell(u) = \nabla_\theta \ell(u)$, i.e. the derivative of the log-likelihood with respect to the neural network parameters.

## 3 PARTIAL TRACE-CLASS BAYESIAN NEURAL NETWORKS

We now propose different neural network structures that contain both Bayesian and non-Bayesian parameters, giving rise to a posterior distribution over the Bayesian parameters, while the non-Bayesian parameters are trained using off-the-shelf optimisation methods. For the Bayesian parameters, we adopt the trace-class prior described in Section 2.1, which gives rise to a natural ordering of the nodes. Due to the split of the parameter space and the involved prior, we coin our architectures *Partial Trace-Class Bayesian Neural Networks* (PaTraC BNNs). The proposed PaTraC BNNs achieve better numerical performance, both in terms of computational speed per inference step and

sample mixing speed, and have a lower memory requirement when compared to full (trace-class) BNNs. Furthermore, full Bayesian inference remains possible on the output function, with the posterior distributions of standard trace-class BNNs and PaTraC BNNs being close. There is a beneficial trade-off in computational efficiency, mixing speed, and memory requirement versus the quality of the uncertainty quantification retained, which we carefully examine in a numerical study in Section 4. In what follows, we introduce the three different PaTraC BNN architectures, each of which has a total of $K$ Bayesian parameters; a table summarising the key properties and differences can be found in Section E in the Supplementary Material.

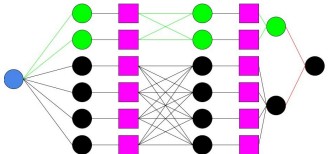 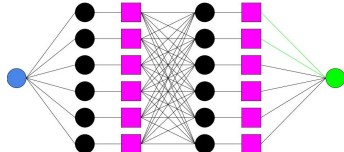 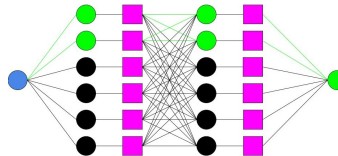

(a) Sep-PaTraC BNN structure with two Bayesian nodes and four optimised nodes in each hidden layer of the NN.

(b) Out-PaTraC BNN structure with six nodes in each hidden layer and two Bayesian weights on the output layer.

(c) Mix-PaTraC BNN structure on a network where we have six nodes in each hidden layer and two Bayesian nodes in each layer.

Figure 1: An illustration of the three different PaTraC BNN structures; green lines denote Bayesian weights, green nodes have associated Bayesian biases, black lines denote optimised weights and black nodes have associated optimised biases. The two red lines at the output in Subfigure 1a denote a fixed weight of $1$ (there is no bias on the very last node in the diagram). Note that we explicitly visualise the non-linear activation function as distinct lines going into the post-activation nodes represented by pink squares.

### 3.1 SEPARATE NETWORKS – SEP-PaTraC BNN

This architecture, illustrated in Figure 1a, consists of a standard neural network running alongside a fully Bayesian neural network. The only interaction between these two is in the output layer, where we sum the respective outputs of the two networks to get the PaTraC BNN output. The overall posterior distribution arises from first training the non-Bayesian neural network, and then using e.g. MCMC for posterior inference on the PaTraC BNN. This PaTraC BNN architecture can be interpreted as estimating the posterior mode using the non-Bayesian neural network, and estimating the uncertainty around it using a smaller but fully Bayesian neural network.

### 3.2 PARTIAL-BAYESIAN OUTPUT LAYER – OUT-PaTraC BNN

This PaTraC BNN architecture, illustrated in Figure 1b, arises from training a neural network before turning those parameters on the output layer into Bayesian weights which correspond to the most important nodes on the last hidden layer. More precisely, let $\eta_i := \left(b_i^{(L-1)}\right)^2 + \sum_{j=1}^{N^{(L-2)}} \left(w_{i,j}^{(L-1)}\right)^2$ be the sum of the squared biases and weights going into the $i$th node of the last hidden layer. The Bayesian parameters are the $N^{(L)}$ biases of the output nodes, as well as the $K/N^{(L)} - 1$ weights for each node on the output layer corresponding to the largest $\eta_i$. To ensure that each output node has the same number of Bayesian weights going into it, $K$ should be a multiple of $N^{(L)}$, which is always the case when there is only a single output node. Note that the out-PaTraC BNN can be derived from Algorithm 1 by setting $k = 0$ in lines 2–21, and $k = K/N^{(L)} - 1$ in lines 22–25.

In a spirit similar to Franssen & Szabó (2022), the interpretation here is that we can decompose each output function into an expansion of 'basis' functions, i.e. the functions on the last hidden layer. The key difference to their work is that we consider a mixture of both optimised and Bayesian weights. Alternatively, the out-PaTraC BNN can also be thought of as equivalent to having a BNN with $N^{(L-1)}$ inputs, $N^{(L)}$ outputs, and no hidden layers, where we choose all the output biases and $K - N^{(L)}$ of the weights to be Bayesian (according to the order defined in the previous paragraph), while the others are kept as the optimised weights trained previously.

### 3.3 MIXED NETWORKS – MIX-PATRAC BNN

This structure, illustrated in Figure 1c, expands the idea of having a partially Bayesian layer, and applies it across all layers of our network. As with the out-PaTraC BNN, a non-Bayesian neural network is first trained, after which the nodes of each layer in the neural network are ordered using the same method described in the previous subsection. After the ordering step, the top $k$ nodes from each layer are selected as Bayesian nodes, i.e. the biases attached to these nodes become Bayesian and any weights which connect two Bayesian nodes together also become Bayesian. Note that a sep-PaTraC BNN with $k$ nodes per layer and a mix-PaTraC BNN with $k$ Bayesian nodes per layer will have the same number of Bayesian parameters. Pseudocode for sampling from the mix-PaTraC BNN prior is given in Algorithm 1.

Notably, the mix-PaTraC BNN is a generalisation of the other two architectures. In comparison to the sep-PaTraC BNN, it allows full interaction between the Bayesian and the optimised neural networks, while when comparing it to the out-PaTraC BNN, Bayesian parameters can be found on all layers of the network.

---

**Algorithm 1** Procedure to sample from the mix-PaTraC BNN prior. In Line 7, the order statistic of the $\eta_i$ is defined, resulting in a relabelling of the nodes where $\mathfrak{i} = 1$ corresponds to the largest $\eta_i$, $\mathfrak{i} = 2$ is the second largest, and so on. Tie-breaking is done using lexicographic ordering of the $i$.

1: **Input**: $k \in \mathbb{N}_0$, $\left( (w_{opt})_{i,j}^{(l)} \right)_{i=1,j=1,l=1}^{N^{(l)}, N^{(l-1)}, L}$, $\left( (b_{opt})_i^{(l)} \right)_{i=1,l=1}^{N^{(l)}, L}$ ▷ Input the number of Bayesian nodes per layer, and the weights and biases after training/optimisation.

2: **for** $l \in [L-1]$ **do**

3:     **for** $i \in [N^{(l)}]$ **do**

4:         $\eta_i := \left( (b_{opt})_i^{(l)} \right)^2 + \sum_{j=1}^{N^{(l-1)}} \left( (w_{opt})_{i,j}^{(l)} \right)^2$

5:     **end for**

6:     **for** $i \in [N^{(l)}]$ **do**

7:         $\mathfrak{i} \leftarrow \sum_{m=1}^{N^{(l)}} \mathbb{1}\{\eta_i \leq \eta_m\}$ ▷ Order statistic of the $\eta_i$.

8:         **if** $\mathfrak{i} \leq k$ **then** ▷ Assign the Bayesian prior if the node is in the largest $k$ nodes.

9:             $b_{\mathfrak{i}}^{(l)} \sim N\left( 0, \frac{\sigma_{b^{(l)}}^2}{\mathfrak{i}^\alpha} \right)$

10:           **if** $l = 1$ **then**

11:              $w_{\mathfrak{i},j}^{(l)} \sim N\left( 0, \frac{\sigma_{w^{(l)}}^2}{\mathfrak{i}^\alpha} \right), \forall j \in [N^{(0)}]$

12:           **else if** $l \neq 1$ **then**

13:              $w_{\mathfrak{i},j}^{(l)} \sim N\left( 0, \frac{\sigma_{w^{(l)}}^2}{(\mathfrak{i}j)^\alpha} \right), \forall j \in [k]; w_{\mathfrak{i},j}^{(l)} \leftarrow (w_{opt})_{i,j}^{(l)}, \forall j \notin [k]$

14:           **end if**

15:         **else** ▷ Small nodes remain non-Bayesian.

16:           $b_{\mathfrak{i}}^{(l)} \leftarrow (b_{opt})_i^{(l)}; w_{\mathfrak{i},j}^{(l)} \leftarrow (w_{opt})_{i,j}^{(l)}, \forall j \in [N^{(l-1)}]$

17:         **end if**

18:     **end for**

19: **end for**

20: **for** $i \in [N^{(L)}]$ **do** ▷ Treat the output layer separately

21:     $b_i^{(L)} \sim N\left( 0, \frac{\sigma_{b^{(L)}}^2}{i^\alpha} \right); \quad w_{i,j}^{(L)} \sim N\left( 0, \frac{\sigma_{w^{(L)}}^2}{(ij)^\alpha} \right), \forall j \in [k]; \quad w_{i,j}^{(L)} \leftarrow (w_{opt})_{i,j}^{(L)}, \forall j \notin [k]$

22: **end for**

23: **Output**: $\left( w_{i,j}^{(l)} \right)_{i=1,j=1,l=1}^{N^{(l)}, N^{(l-1)}, L}$, $\left( b_i^{(l)} \right)_{i=1,l=1}^{N^{(l)}, L}$ ▷ Return a prior sample, containing all weights and biases.

---

## 4 NUMERICAL SIMULATION STUDY

We now aim to compare our different PaTraC BNN architectures to each other, and to the full Bayesian neural network, as well as to optimised neural networks. In particular, in Subsection 4.2

we show the ability of PaTraC BNNs to recover meaningful uncertainty quantification, in 4.3 we highlight the improved mixing behaviour of the PaTraC BNNs' Markov chains when compared to full BNNs, and analyse the computational and statistical efficiency of our proposed approaches. These insights allow us to argue in Section 6 that there exists a trade-off between obtaining good uncertainty quantification, lowering computational cost, and increasing mixing speeds. Subsection 4.1 describes our experimental setup in detail; all code was run on a laptop with 16 GB RAM and an i7 Intel Core, without the use of GPUs.

## 4.1 STUDY SETUP

We assess our methodology on a simple toy training data set consisting of 100 data points $(x_i, y_i)_{i=1}^{100}$, where $x_i \sim \mathcal{U}[-5, 5]$, and $y \sim N(f(x), 1)$ with $f(x) = \sin(x)$. We sample a further $1,000$ test data points from the same distribution for model comparison and evaluation. To assess the robustness with respect to the training set, we will be comparing the performance across 100 experiments with independently generated training and test data.

Our model supposes that there exists a neural network indexed by parameters $\theta$ such that $f_\theta \approx f$. Throughout this section we use feed-forward fully-connected neural networks with 2 hidden layers of 50 nodes, 1 input, and 1 output node, which are connected by $\tanh$ activation functions. We train neural networks using Adam (Kingma, 2014) with a learning rate of $10^{-6}$ and an $L^2$ penalty of 1, the loss function is the negative log-likelihood. Any networks were trained until there were 20 consecutive steps of non-improvement on the loss.

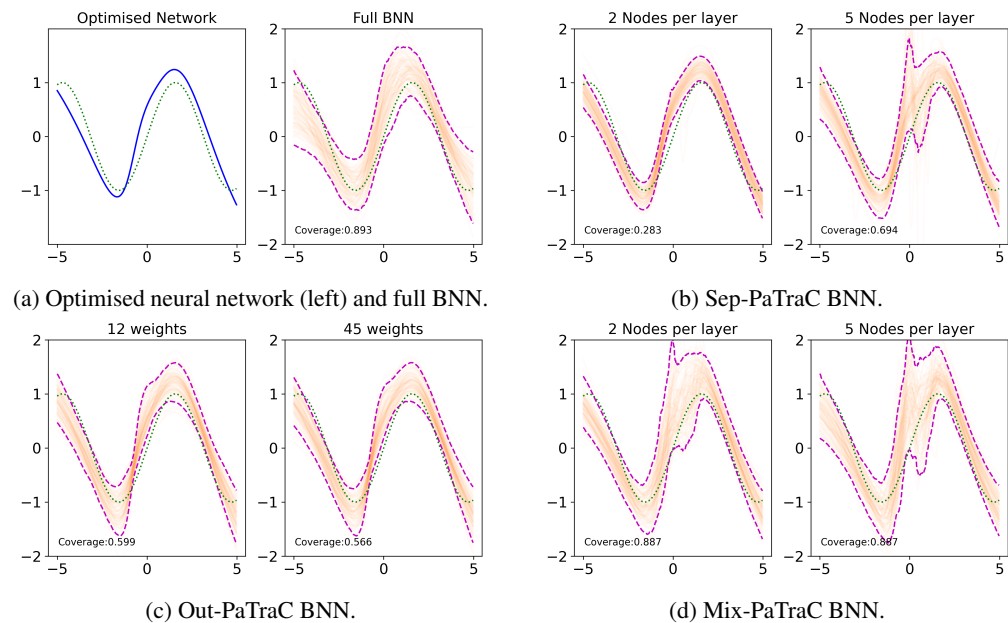

Figure 2: Plots of the different posterior distributions, shown are 100 samples from the respective posteriors (orange). The green dotted lines indicate the true mean function we are attempting to predict, the dashed purple lines are the respective $2.5\%$ and $97.5\%$ posterior quantiles.

The fully Bayesian version of this neural network, as well as the PaTraC BNNs, is equipped with the trace-class prior described in Section 2.1. For the PaTraC BNNs, we have two different models, with differing numbers of Bayesian nodes. For both the sep-PaTraC BNN and mix-PaTraC BNN, we chose $k \in \{2, 5\}$ Bayesian nodes per layer, for the out-PaTraC BNN we chose $k \in \{12, 45\}$ Bayesian weights on the output layer. This results in a total number of 13 and 46 Bayesian weights for each of the different architectures, respectively. The hyperparameter choices are detailed in Section A in the Supplementary Material.

Our burn-in procedure is outlined in Section B in the Supplementary Material. After burn-in, we use an adaptive pCNL tuning parameter $\delta$ to ensure that the acceptance probability falls into the interval $(0.4, 0.9)$, see Section A in the Supplementary Material for details on the adaptive choice of

$\delta$. $500,000$ samples are collected and thinned to $500$ effective samples used in the results presented below.

## 4.2 COMPARISON OF UNCERTAINTY QUANTIFICATION

Figure 2 shows how the PaTraC BNN posteriors compare to a full BNN, based on a fixed training data set. For each of the PaTraC BNN architectures, we picked two different numbers of Bayesian nodes to illustrate the effect of this quantity on the quality of approximation to the full BNN. The first plot in the top row shows the function $f_{\theta_{\text{opt}}}$, i.e. the optimised neural network regression function. The orange functions are samples from the respective posteriors. We see that, even with few Bayesian parameters, the PaTraC posterior samples are relatively close to the optimised function, this effect decreases as more Bayesian parameters are included, when the PaTraC posteriors get closer to the full BNN posterior.

We analyse the ability to capture uncertainty over 100 independent repetitions of the above experiment, generating new train and test sets for each of the runs. In Figure 3, we report boxplots of the observed coverages, that is, $\frac{1}{1000} \sum_{i=1}^{1000} \mathbb{1}\{f(x_i^{\text{test}}) \in [\hat{\pi}_{\tau/2}(f_\theta(x_i^{\text{test}})), \hat{\pi}_{1-\tau/2}(f_\theta(x_i^{\text{test}}))]\}$ for $\tau \in \{0.01, 0.05, 0.35\}$, and where $\hat{\pi}_\beta(f_\theta(x)) := \text{argmin}\{q \in \mathbb{R} : \frac{1}{M}\sum_{j=1}^{M} \mathbb{1}\{f_{\theta_j}(x) < q\} \leq \beta\}$ is an estimator of the $\beta$-quantile based on $M = 500$ posterior samples at a fixed point $x$, and $f(x) = \sin(x)$ is the true function of interest. The mix-PaTraC BNN appears to achieve coverages similar to the full BNN, while the out-PaTraC BNN and sep-PaTraC BNN show slightly worse coverages, with more Bayesian parameters improving the coverage for all architectures.

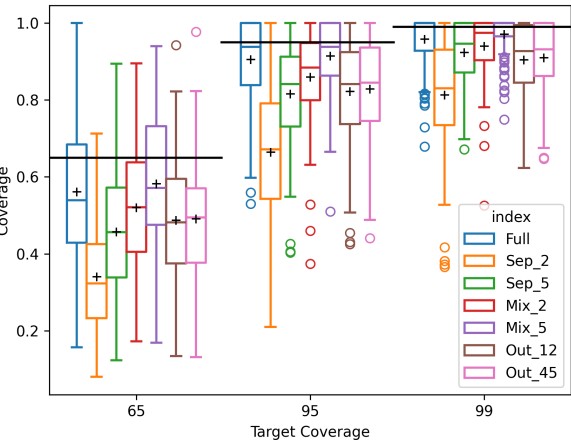

Figure 3: Box plots of observed coverage for the different architectures across 100 experiments. Black lines: target coverages of $65\%$, $95\%$, and $99\%$, respectively.

Table 1: Empirical predictive negative log-likelihood values, averaged over 100 independent runs (1000 test points per run), with standard deviations across the different runs shown in brackets.

| Architecture | NLL |
|---|---|
| Full | $571.775_{(31.639)}$ |
| Sep_2 | $547.724_{(17.857)}$ |
| Sep_5 | $557.431_{(24.865)}$ |
| Mix_2 | $567.272_{(30.018)}$ |
| Mix_5 | $575.944_{(33.628)}$ |
| Out_12 | $559.644_{(24.931)}$ |
| Out_45 | $559.646_{(25.388)}$ |
| Laplace | $576.478_{(60.180)}$ |

## 4.3 MIXING SPEED AND COMPUTATION TIME

Our main motivation for proposing PaTraC BNNs was an expected speed-up in terms of both posterior mixing and computational cost. Traceplots, illustrating the mixing behaviour, are presented in Section D in the Supplementary Material. Table 2 shows the effective sample size (ESS) and ESS per second (ESS/s) for the different architectures.

Considering the different architectures, we first observe that the mix-PaTraC BNNs offer hardly any benefit in terms of ESS/s in comparison to the full BNN, although we suspect this is due to our code not being optimised for efficient computation. Both the sep-PaTraC BNN and the out-PaTraC BNN offer a significant speed up. Interestingly, the ESS/s increases with the number of Bayesian nodes for both the mix-PaTraC BNN and the out-PaTraC BNN, which is the opposite of what we would have expected. We conjecture this is due our ESS calculation focusing on the target space of interest, rather than the parameter space, see Section C in the Supplementary Material for the implementation details of our ESS measure.

Table 2: Effective sample size (ESS) per second, ESS, and computation time (in seconds), all averaged over 100 independent experiment repetitions. The standard deviations across these runs are reported in brackets.

| BNN | Mean ESS/s | Mean ESS | Mean time |
|---|---|---|---|
| Full | $3.7618_{(1.6740)}$ | $2591.23_{(1150.48)}$ | $689.187_{(8.544)}$ |
| Sep_2 | $23.6800_{(6.1666)}$ | $9787.71_{(2530.85)}$ | $413.467_{(5.979)}$ |
| Sep_5 | $14.6011_{(4.2347)}$ | $6103.98_{(1757.65)}$ | $418.197_{(5.886)}$ |
| Mix_2 | $3.0764_{(1.2894)}$ | $1692.30_{(704.76)}$ | $550.512_{(9.072)}$ |
| Mix_5 | $4.0604_{(1.0869)}$ | $2244.97_{(600.24)}$ | $552.933_{(7.171)}$ |
| Out_12 | $12.0872_{(13.7228)}$ | $3073.65_{(3492.16)}$ | $254.567_{(4.955)}$ |
| Out_45 | $13.3810_{(13.2031)}$ | $3404.58_{(3362.53)}$ | $254.689_{(4.547)}$ |

## 5 REAL DATA EXAMPLES

### 5.1 CIFAR-10

To verify the scalability of the different PaTraC BNN architectures to more challenging examples, we compare performance and computation time on the CIFAR-10 image classification dataset (Krizhevsky, 2009). The log-likelihood was chosen to be the negative cross-entropy loss scaled by a factor of $1,000$ to account for the informativeness of the chosen prior. The results are presented in Table 3. The full BNN takes longest to be sampled from, closely followed by the mix-PaTraC BNN due to it requiring full gradient calculations. The sep-PaTraC and out-PaTraC are magnitudes faster. We also observe that the negative log-likelihood (NLL) of these PaTraCs is much closer to the NLL of the optimised NN, which resembles what was observed in the numerical study in Section 4. Finally, we note that we cannot assess the usefulness of the uncertainty quantification in this real data example in the same way as was done in Section 4.

### 5.2 ABALONE

To illustrate the performance of our proposed methodology on real data, we use the abalone data set (Nash et al., 1994), aiming to predict the number of rings (proportional to the age) of abalones based on 8 features. We split the data into $N = 2,923$ training and $N_{\text{test}} = 1,254$ test points. The study was conducted using the same architectures and priors as in Section 4, with the same training and sampling procedures (except for changing the $L^2$ penalty to 0.01), and on the same computer. The likelihood is chosen to be normal with mean $f_\theta(x)$ and variance $\sigma^2 = 36$. The results based on 500 posterior samples for each of the different PaTraC BNNs are presented in Figure 4. Apart from the sep-PaTraC BNNs that appear to be overly confident, all the architectures yield reasonable uncertainty quantification. The full BNN gives the best predictive posterior, while the sep-PaTraC-BNNs give the worst and are underestimating the uncertainty around the prediction. As expected, a higher number of Bayesian parameters results in posteriors closer to the full BNNs, in fact, the out-PaTraC BNN with 45 Bayesian

Table 3: Predicted negative log-likelihood (NLL) values for the CIFAR-10 dataset. The standard deviations based on the posterior samples for the full BNN and respective PaTraC architectures are reported in brackets; the NLL for the optimised (non-Bayesian) NN are shown in the first row. The final column reports the time (in seconds) elapsed to obtain 500 thinned samples.

| Architecture | PNLL | Sampling time |
|---|---|---|
| Optimised | 1.3028 | N/a |
| Full | $2.0850_{(0.02799)}$ | 26191.72 |
| Sep_4 | $1.3399_{(0.03118)}$ | 4830.57 |
| Sep_10 | $1.3382_{(0.0311)}$ | 5296.12 |
| Mix_4 | $1.7736_{(0.1361)}$ | 25583.62 |
| Mix_10 | $1.9595_{(0.1855)}$ | 24922.24 |
| Out_24 | $1.3728_{(0.02395)}$ | 475.07 |
| Out_90 | $1.5392_{(0.02672)}$ | 471.10 |

parameters and the mix-PaTraC BNN with 5 Bayesian nodes per layer show behaviour virtually undistinguishable from the full BNN.

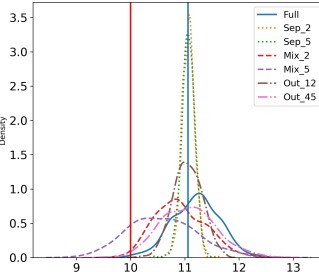 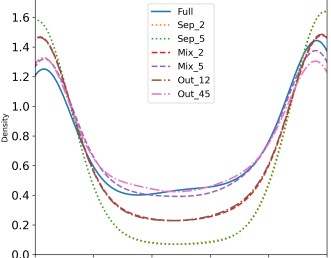

Figure 4: Results from the abalone data set. Left: for a single test point, the true number of rings is shown as a red vertical line, the prediction of the trained neural network is shown as a blue vertical line. The posterior predictive distributions are shown for the full BNN (blue line), sep-PaTraC (orange and green dotted lines), mix-PaTrac (red and purple dashed line), and out-PaTraC (brown and pink dotted-dashed line). Right: kernel density estimates for posterior quality comparison, shown is the empirical distribution of $\sum_{i=1}^{500} \mathbb{1}\{f_{\theta_i}(x) < y\}/500$ over all $(x, y)$ in the test set. Distributions close to uniform correspond to better posterior quality, while convex and concave distributions correspond to over- and under-confident posteriors, respectively.

## 6 DISCUSSION

We proposed three new Bayesian neural network architectures that are more computationally and statistically efficient than full Bayesian neural networks by reducing the number of Bayesian parameters based on the relative ordering of weights from a trained neural network. Similar approaches have been explored by Izmailov et al. (2020); Daxberger et al. (2021b); Calvo-Ordoñez et al. (2024), with our work differing in two key ways: first, we use trace-class BNN priors (Sell & Singh, 2023) which inherently introduce an order of the neural network parameters, and naturally suggest a node-based procedure to select Bayesian parameters. Second, we select *nodes* based on their relative importance after optimising the network parameters using an off-the-shelf optimiser, and turn the associated weights into Bayesian parameters. This is in contrast to existing approaches that select *parameters* on their respective values after training, choose full layers to be Bayesian, or focus only on a subset of the final layer weights to be Bayesian.

Our results presented in Section 4.2 show that the three different architectures provide uncertainty quantification comparable to the full Bayesian neural network, while only using a fraction of the number of Bayesian parameters. Between the three different architectures, the mix-PaTraC BNN appears to be most similar to the full BNN, while the sep-PaTraC BNN is over-confident in comparison to the other architectures, particularly if few Bayesian parameters are used. In Section 4.3, we observed that the sep-PaTraC BNN offers the highest ESS/s in comparison to the other architectures, while the mix-PaTraC BNN showed only a marginal improvement in terms of ESS/s. This lack of ESS/s improvement, despite a reduced number of Bayesian parameters, is discouraging on first sight, but we conjecture this can be explained by the fact that these target different posteriors, and by the fact that we did not optimise our code in any way, e. g. by optimising GPU use and avoiding unnecessary computation in the gradient calculations. This computational bottleneck thus holds the potential to significantly improve the relative performance of all PaTraC BNNs compared to the full BNN.

On the positive side, the reduction in computation and memory requirements that a PaTraC BNN can provide would lessen the negative impact of large-scale machine learning on the environment. Our methodology thus addresses the concerning negative environmental impact of large machine learning models, one of the main ethical considerations in AI.

Aside from computational considerations, a theoretical investigation into the properties of the PaTraC BNNs in the infinite width limit would strengthen the foundations of the proposed methodology, in particular if the Wasserstein distance of the induced posteriors on the output space can be explicitly bounded.

## 7 REPRODUCIBILITY STATEMENT

The architectures used are clearly described in Section 3, with specific parameter choices detailed in Section A in the Appendix. Algorithm 1 describes in pseudo code the procedure to sample from the mix-PaTraC prior. All details to reproduce our numerical simulations are provided at the beginning of Section 4 with some details outsourced to the Appendix, see Sections B and C. A GitHub repository containing all code will be made public upon publication.

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

## APPENDIX

## A   HYPERPARAMETER CHOICE

Recall that we have a set of optimised parameters which we denote as $\hat{\theta}$. First, we define

$$(\alpha, \sigma_w^2/4, \sigma_b^2/4) := \underset{\alpha, \sigma_w^2, \sigma_b^2}{\operatorname{argmin}} \left( \sum_{l \in [L]} \sum_{i \in [N^{(l)}]} \left\{ D_b(l, i, \sigma_b^2) + \sum_{j \in [N^{(l-1)}]} D_w(l, i, j, \sigma_w^2) \right\} \right),$$

where

$$D_b(l, i, \sigma_{b^{(l)}}^2) = \left( (\hat{b}_i^{(l)})^2 - \frac{\sigma_{b^{(l)}}^2}{i^\alpha} \right)^2, \text{ for } 1 \leq l \leq L,$$

$$D_w(l, i, j, \sigma_{w^{(l)}}^2) = \left( (\hat{w}_{i,j}^{(l)})^2 - \frac{\sigma_{w^{(l)}}^2}{(ij)^\alpha} \right)^2, \text{ for } 2 \leq l \leq L - 1,$$

$$D_w(1, i, j, \sigma_{w^{(1)}}^2) = \left( (\hat{w}_{i,j}^{(1)})^2 - \frac{\sigma_{w^{(1)}}^2}{i^\alpha} \right)^2,$$

$$D_w(L, i, j, \sigma_{w^{(L)}}^2) = \left( (\hat{w}_{i,j}^{(L)})^2 - \frac{\sigma_{w^{(L)}}^2}{j^\alpha} \right)^2.$$

Now, the remaining hyperparameters for the full BNN are chosen as

$$\sigma_{w^{(1)}}^2 = \sigma_{w^{(2)}}^2 = \cdots = \sigma_{w^{(L)}}^2 := \sigma_w^2, \text{ and } \sigma_{b^{(1)}}^2 = \sigma_{b^{(2)}}^2 = \cdots = \sigma_{b^{(L)}}^2 := \sigma_b^2.$$

Furthermore, for the PaTraC BNNs, we scale all parameters to match the overall prior variance to match the variance of the full BNN, similar to Daxberger et al. (2021b). To this end, we define the total weight and total bias variances of the full BNN as

$$\phi_w = \sum_{l=1}^{L} \sum_{i=1}^{N^{(l)}} \sum_{j=1}^{N^{(l-1)}} Var(w_{i,j}^{(l)}), \qquad \text{and} \qquad \phi_b = \sum_{l=1}^{L} \sum_{i=1}^{N^{(l)}} Var(b_i^{(l)}),$$

respectively, and let $\phi := \phi_w + \phi_b$. For the PaTraC BNNs, similar quantities are computed based on all Bayesian parameters in the respective PaTraC BNN, which we denote $\phi^{\text{PaTraC}}$. We then rescale the PaTraC BNN prior as

$$w_{i,j}^{(1)} \sim N\left( 0, \frac{\sigma_{w^{(1)}}^2}{i^\alpha} \frac{\phi}{\phi^{\text{PaTraC}}} \right), \qquad w_{i,j}^{(l)} \sim N\left( 0, \frac{\sigma_{w^{(l)}}^2}{(ij)^\alpha} \frac{\phi}{\phi^{\text{PaTraC}}} \right) \quad \text{for } l \geq 2,$$

$$b_i^{(l)} \sim N\left( 0, \frac{\sigma_{b^{(l)}}^2}{i^\alpha} \frac{\phi}{\phi^{\text{PaTraC}}} \right).$$

## B   BURN-IN PROCEDURE

We will make use of pCNL with an adaptive tuning parameter $\delta$ to ensure that the acceptance probability falls into a pre-specified interval $(p_L, p_U)$. To this end, let $\hat{p}_{\text{acc}}$ denote the relative frequency of acceptance in the last 200 samples. If $\hat{p}_{\text{acc}} \leq p_l$ then update $\delta \leftarrow \frac{\delta}{2}$, if $\hat{p}_{\text{acc}} \geq p_u$ then update $\delta \leftarrow \min\{\frac{4\delta}{3}, 2\}$, otherwise leave $\delta$ unchanged.

While other burn-in procedures are possible, we found the following procedure to lead to a quick burn-in:

1. Initialise $\delta := 10^{-4}$.
2. Run $50,000$ pCNL steps with an adaptive delta and $(p_L, p_U) := (0.85, 0.95)$.
3. Run $100,000$ pCNL steps with every move accepted, and do not adjust $\delta$.
4. Run another $500,000$ pCNL steps with adaptive $\delta$ and $(p_L, p_U) := (0.4, 0.9)$.

## C    EFFECTIVE SAMPLE SIZE CALCULATION

We are ultimately interested in the induced posterior on the joint input and output space, and are thus estimating the effective sample based on an evaluation metric based on a regular grid on the input space. To this end, let $R = \{r_1, r_2, \ldots, r_g\}$ be a grid of inputs to the BNN and let $\Theta = \{\theta^{(1)}, \ldots, \theta^{(N)}\}$ be $N$ posterior samples on the parameter space. For $k \in [1, 1000]$, let

$$\gamma_k = \frac{1}{N-k-1} \sum_{n=1}^{N-k} \frac{1}{|R|} \sum_{r \in R} \left( (f_{\theta^{(n)}}(r) - \mu(r))(f_{\theta^{(n+k)}}(r) - \mu(r)) \right),$$

where $\mu(x) = \frac{1}{n} \sum_{n=1}^{N} f_{\theta^{(n)}}(x)$. Considering the vector

$$(f_\theta(R) - \mu(R)) = [(f_\theta(r_1) - \mu(r_1)), \ldots, (f_\theta(r_g) - \mu(r_g))]^T,$$

we note that

$$\gamma_k = \frac{1}{N-k-1} \sum_{n=1}^{N-k} \frac{1}{|R|} \left( (f_{\theta^{(n)}}(R) - \mu(R))^T (f_{\theta^{(n+k)}}(R) - \mu(R)) \right)$$

$$= \frac{1}{|R|} \sum_{r \in R} \frac{1}{N-k-1} \sum_{n=1}^{N-k} \left( (f_{\theta^{(n)}}(r) - \mu(r))(f_{\theta^{(n+k)}}(r) - \mu(r)) \right),$$

i. e. $\gamma_k$ is an estimator of the mean autocovariance of $f_{\theta^{(n)}}(r_m)$ across the $|R|$ chosen grid points. This allows us to obtain an estimator of the corresponding mean autocorrelations by letting

$$\hat{\rho}_k := \frac{1}{|R|} \sum_{r \in R} \frac{1}{N-k-1} \frac{1}{\hat{\sigma}_r^2} \sum_{n=1}^{N-k} \left( (f_{\theta^{(n)}}(r) - \mu(r))(f_{\theta^{(n+k)}}(r) - \mu(r)) \right)$$

where $\hat{\sigma}_r^2 = \frac{1}{N-1} \sum_{n=1}^{N} (f_{\theta^{(n)}}(r) - \mu(r))^2$. We can then define an estimator of the effective sample size as usual by

$$\widehat{ESS} := \frac{N}{-1 + 2 \sum_{k=0}^{1000} \hat{\rho}_k},$$

where we set $\hat{\rho}_0 = 1$.

We note that other notions of effective sample size may be of interest, see e. g. Papamarkou et al. (2022), however, the differences observed compared to the above results were not significant enough to change the interpretation of them.

## D    TRACEPLOTS

Tables 4 to 7 show traceplots to visually assess mixing behaviour of the MCMC algorithm for the different posterior distributions arising from the full BNN and the three PaTraC BNNs with different parameters. Displayed are traceplots for the log-likelihood, log-prior, the first weight on the last layer, and the output function evaluated at $x = -2$ (mean centred).

Table 4: Table of traceplot figures for the full BNN, and the Sep_2 and Sep_5 PaTraC BNNs. Shown are the traces of the log-likelihood and log-prior for $500,000$ samples. The table is continued on the next page.

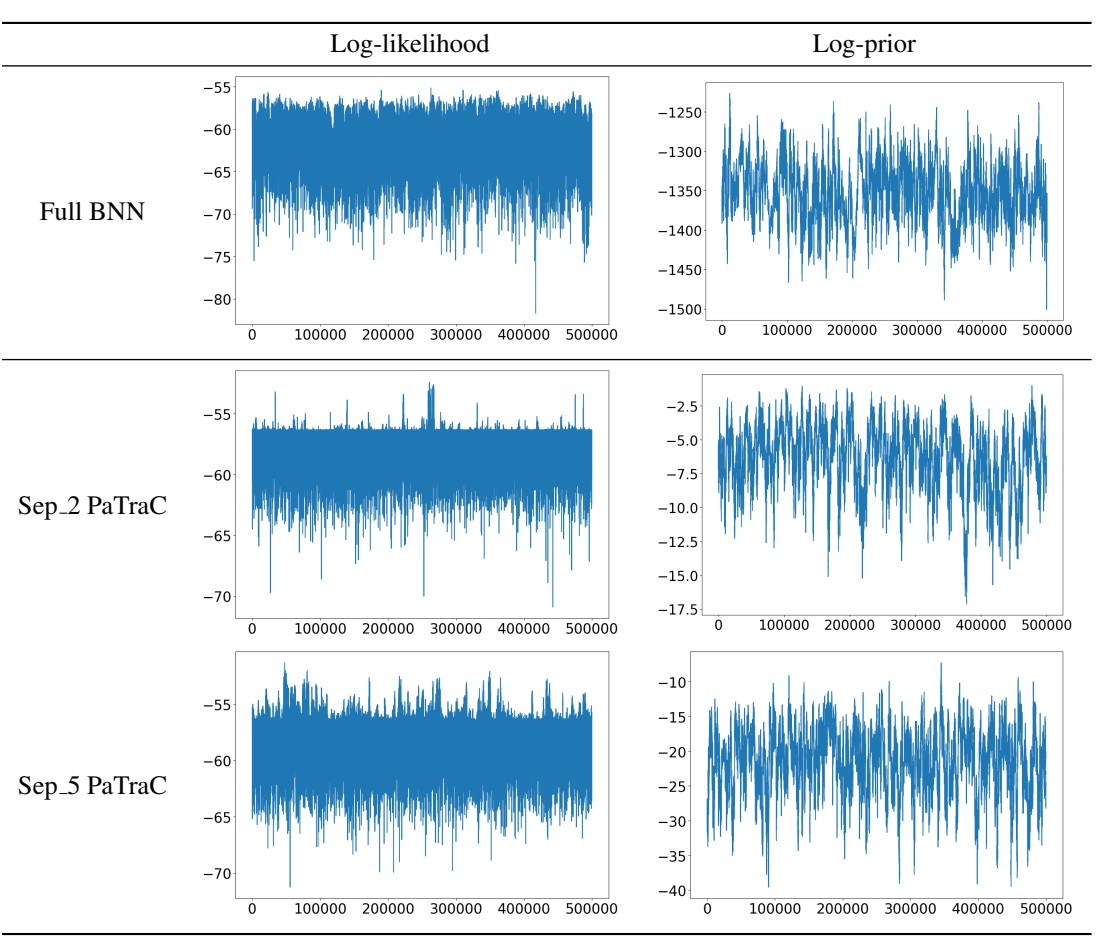

Table 5: Table (continued) of traceplot figures for the full BNN, and the Sep_2 and Sep_5 PaTraC BNNs. Shown are the traces of the first weight on the last layer, and the output function evaluated at $x = -2$ for $500,000$ samples.

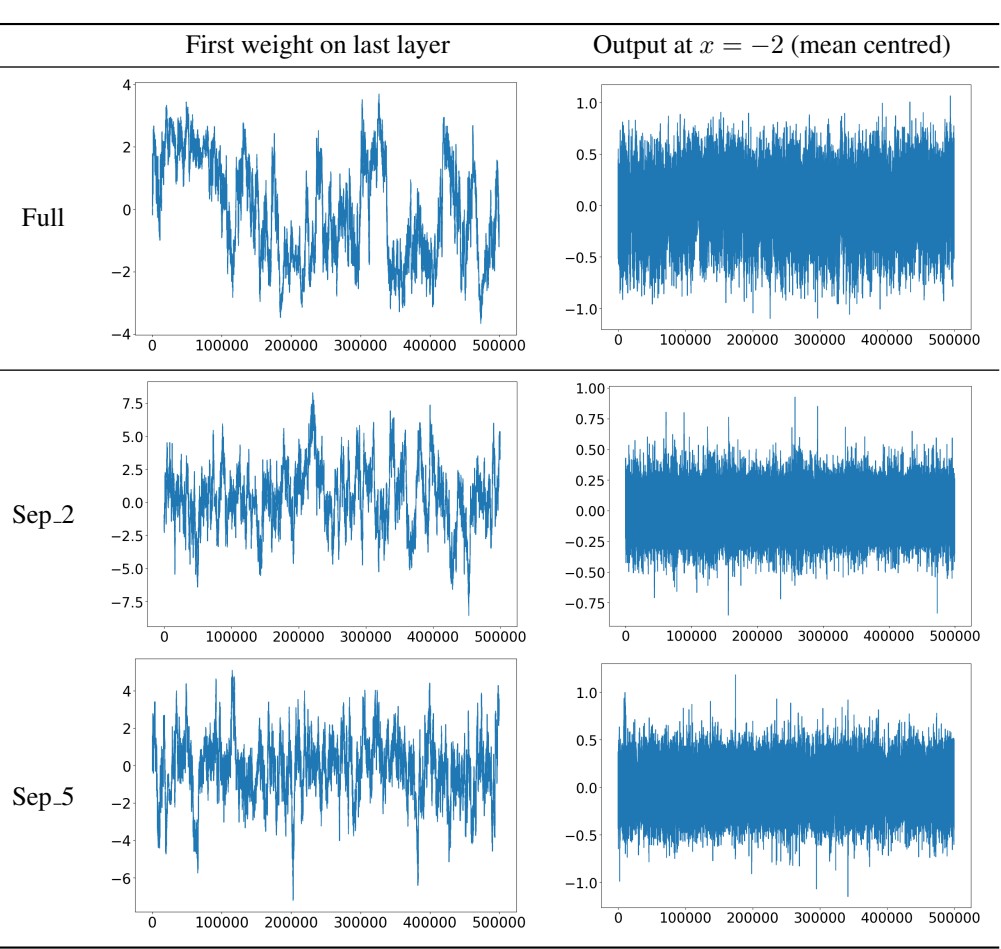

Table 6: Table of traceplot figures for the Out_12, Out_45, Mix_2, and Mix_5 PaTraC BNNs. Shown are the traces of the log-likelihood and log-prior for $500,000$ samples. The table is continued on the next page.

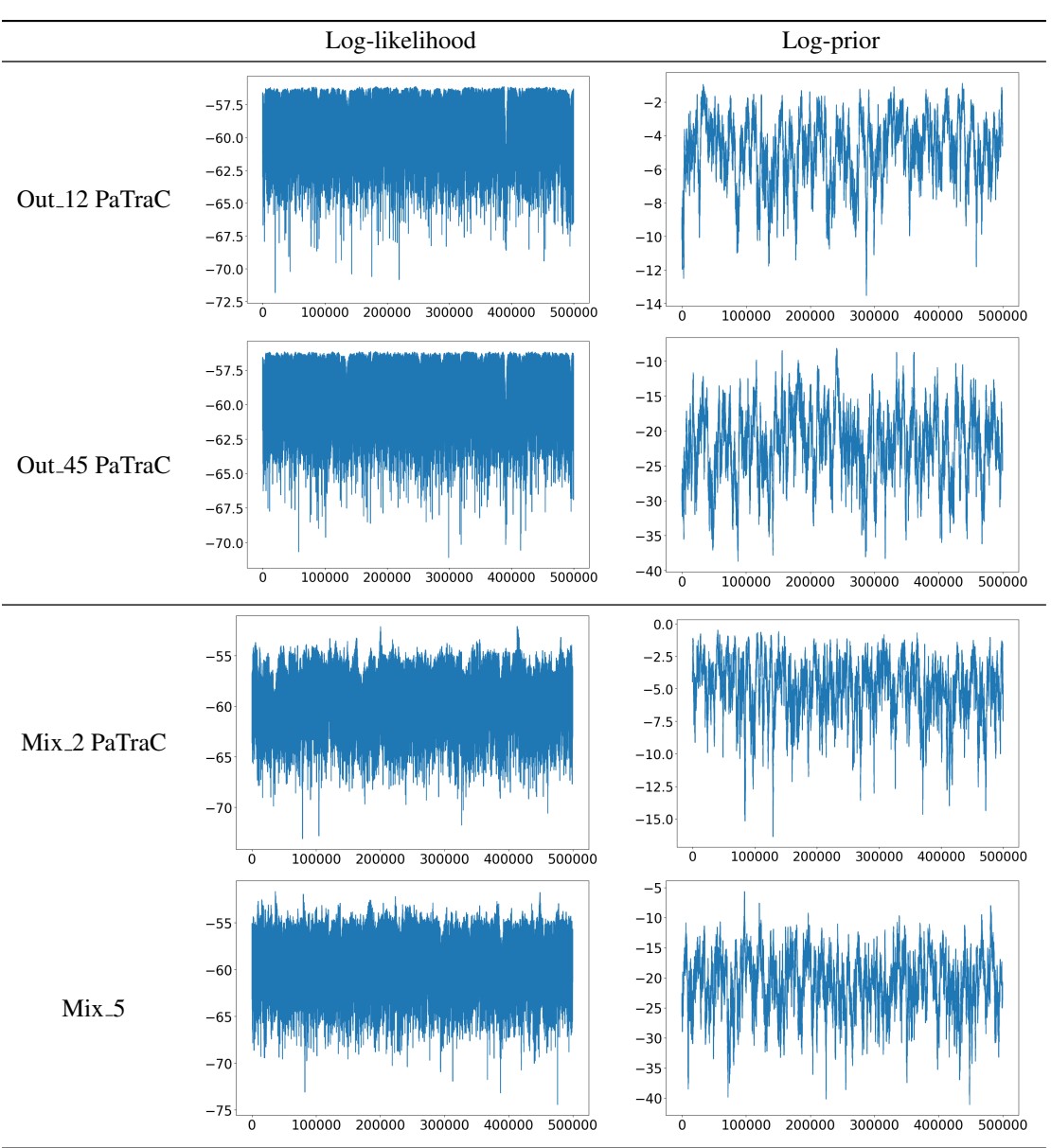

Table 7: Table (continued) of traceplot figures for the Out_12, Out_45, Mix_2, and Mix_5 PaTraC BNNs. Shown are the traces of the first weight on the last layer, and the output function evaluated at $x = -2$ for $500,000$ samples.

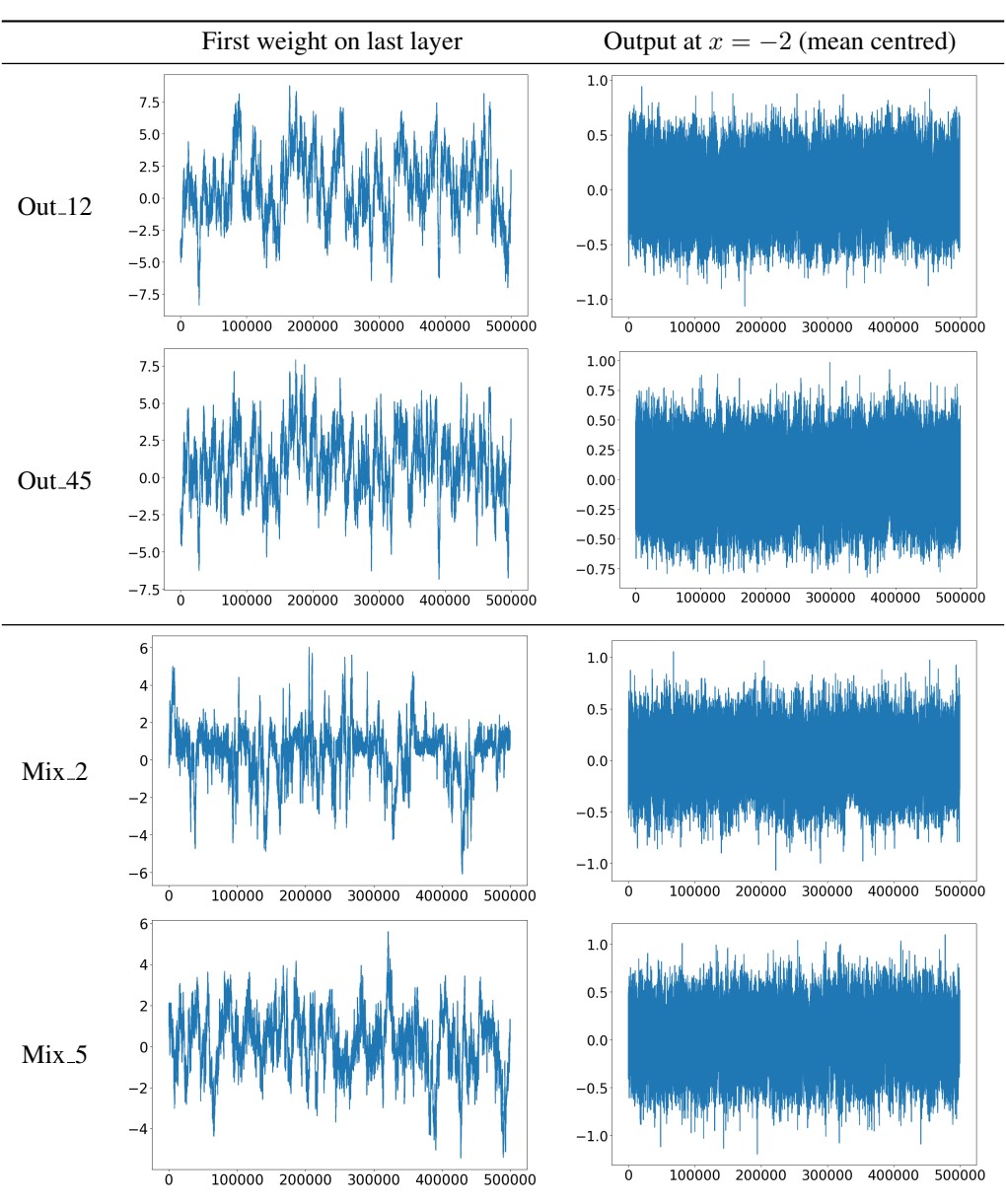

