# OpenReview forum: "Partial Trace-Class Bayesian Neural Networks"
_ICLR.cc/2026/Conference — ICLR 2026 Conference Withdrawn Submission_

### Official Review · Reviewer_WBbu · 2025-10-29

**Soundness:** 1
**Presentation:** 2
**Contribution:** 1
**Rating:** 2
**Confidence:** 5

**Summary:**

The paper introduces novel Bayesian neural network (BNN) architectures that enable scalable and reliable uncertainty quantification with reduced computational cost. The authors propose three approaches (Sep-PaTraC, Out-PaTraC, and Mix-PaTraC) that integrate Bayesian and non-Bayesian parameters. Their method first trains a standard neural network, then applies a trace-class prior to the most important nodes to capture uncertainty. Posterior inference is performed using a Hilbert space MCMC method. The authors evaluate proposed architectures on both synthetic and real-world datasets.

**Strengths:**

1. The paper addresses the important trade-off between reliable uncertainty quantification and scalable Bayesian inference. It introduces three novel architectures (Sep-PaTraC, Out-PaTraC, and Mix-PaTraC) that effectively combine Bayesian and non-Bayesian components to achieve efficient uncertainty modeling.

2. The authors provide comprehensive empirical validation through experiments on both synthetic data and two real-world datasets (CIFAR-10 and Abalone), demonstrating the versatility of the proposed methods.

3. Computational efficiency is assessed using effective sample size (ESS) and runtime. The Sep-PaTraC and Out-PaTraC architectures, in particular, show practical gains in computation efficiency.

**Weaknesses:**

1. My main concern lies in the inference design, which may undermine the key advantage of using the trace-class prior. According to the authors, the trace-class prior is intended to “introduce a natural ordering of the prior weights and thus naturally lend them to truncation.” However, during pre-training, the standard neural network is trained without any prior information, and therefore, nodes with large ordering statistics may emerge in later layers. After Bayesian inference, these nodes might have posterior means close to zero and contribute little to the final output. It would strengthen the work to investigate whether the selected nodes retain their relative importance after posterior inference and to justify the current inference procedure.

2. The paper lacks empirical justification for adopting the trace-class prior. It would strengthen the work to include baselines using standard Gaussian weight priors to illustrate the specific benefits of the trace-class formulation.

3. The experimental section could be more comprehensive. More specifically,
    * The study tests only one architecture (a feedforward neural network with two hidden layers of 50 nodes) across all three tasks. To demonstrate the generality and robustness of the proposed approach, additional architectures of varying depth and width should be evaluated. Moreover, prior work ([2]) suggests that trace-class priors are particularly advantageous for wide Bayesian neural networks, as they yield well-defined function-space priors in the infinite-width limit. Evaluating the proposed method in this broader regime would provide stronger empirical support for its benefits.
    * For the CIFAR-10 and Abalone experiments, only the negative log-likelihood (NLL) is reported. NLL alone does not guarantee well-calibrated uncertainty estimates (see, e.g., [1]). Including additional metrics such as the expected calibration error (ECE) or evaluating performance under out-of-distribution (OOD) conditions would provide a more complete assessment of uncertainty calibration.
    * Across the presented results, the Mix-PaTraC architecture performs comparably to the full BNN in predictive accuracy, but its computational cost (Tables 2 and 3) is also similar. This raises concerns about the practical advantage of Mix-PaTraC, given that it uses substantially fewer Bayesian parameters. A more detailed discussion of this trade-off would be helpful.

**Reference**

[1] Wang, Cheng. "Calibration in deep learning: A survey of the state-of-the-art." arXiv preprint arXiv:2308.01222 (2023).
[2] Sell, Torben, and Sumeetpal Sidhu Singh. "Trace-class Gaussian priors for Bayesian learning of neural networks with MCMC." Journal of the Royal Statistical Society Series B: Statistical Methodology 85.1 (2023): 46-66.

**Questions:**

1. In Figure 2, although the Mix-PaTraC BNN achieves coverage scores comparable to the full BNN, the posterior samples exhibit noticeable kinks or irregularities, which appear somewhat pathological. Could the authors comment on this behavior or provide an explanation for these artifacts?
2. How rapidly do the prior weights decay across deeper layers under the trace-class prior?

---

### Official Review · Reviewer_sLGG · 2025-10-29

**Soundness:** 2
**Presentation:** 2
**Contribution:** 2
**Rating:** 2
**Confidence:** 4

**Summary:**

The paper proposes three related Bayesian neural network architectures, termed **PaTraC BNNs**, which aim to achieve uncertainty quantification similar to a full Bayesian network while using far fewer Bayesian parameters by leveraging a trace-class prior mechanism inspired by Sell & Singh, 2023. In the trace‑class prior, the variances of the weights and biases decrease with the index of the node, inducing a natural ordering of nodes and enabling truncation of less important parameters. The authors leverage this ordering to create BNNs in which only a subset of network parameters is treated in a Bayesian manner; the remaining parameters are trained via standard deterministic optimization. The three variants are:

1.	**Sep-PaTraC BNN:** A full deterministic network runs in parallel with a smaller fully-Bayesian network; their outputs are summed to form the final prediction.

2.	**Out-PaTraC BNN:** After training a standard network, a subset of the last-layer weights (those connected to the highest-variance hidden nodes) and output biases are converted to Bayesian parameters, while the remaining weights stay fixed.

3.	**Mix-PaTraC BNN:** Extends the partial-Bayesian idea to all layers: it selects the top-$k$ nodes in each layer (based on post-training weight magnitudes) as Bayesian, including their incoming and outgoing weights. The rest of the network remains deterministic.

Posterior inference for the Bayesian parameters is performed with the preconditioned Crank–Nicholson Langevin (pCNL) MCMC algorithm. The authors conduct a simulation study on a toy regression problem (sine function) where PaTraC BNN variants are compared to a full-trace class BNN and to non-Bayesian neural networks. They evaluate the posterior function samples, coverage of posterior credible intervals, and negative log‑likelihoods. The study also measures mixing speed to assess computational efficiency. Additionally, real‑data experiments are presented. The paper concludes that PaTraC BNNs can approximate full BNNs’ uncertainty quantification with significantly reduced computational cost.

**Strengths:**

1.	**Interesting combination of ideas:** The paper creatively blends trace-class priors with partial Bayesian inference. Leveraging the inherent node ordering of the trace prior to guide parameter selection is an interesting strategy.

2.	**Clear definition of architectures:** The three proposed PaTraC variants are described systematically (with figures), making it easy to distinguish their differences and intended use-cases.

3.	**Empirical investigation of trade-offs:** The experiments attempt to quantify the balance between uncertainty quality and computational efficiency. Plotting coverage vs. number of Bayesian parameters, and reporting ESS/time, provides insight into the trade-offs.

4.	**Empirical evaluation of uncertainty quantification.** The simulation study evaluates coverage of posterior credible intervals, predictive negative log‑likelihoods, and effective sample size per second across 100 independent runs. Results show that the mixed architecture’s coverage is comparable to a full BNN, while separate and output‑layer architectures are slightly over‑confident; increasing the number of Bayesian parameters improves coverage.

5.	**Reduction in sampling time on real data.** On CIFAR‑10, the sep‑PaTraC and out‑PaTraC models require an order of magnitude less sampling time than a full BNN while maintaining predictive negative log‑likelihoods close to those of a deterministic neural network. Similar trends are observed on the Abalone dataset.

**Weaknesses:**

1.	**Limited novelty relative to existing pBNN literature.** Prior works on partially Bayesian or subnetwork BNNs (e.g., Daxberger et al., 2021; Izmailov et al., 2020) already propose training deterministic networks and then sampling Bayesian parameters on subsets of layers. The main difference here is using the trace‑class prior’s ordering to choose which nodes to Bayesianize. There is no theoretical analysis demonstrating that this selection method yields superior posterior approximations, and the idea of ordering nodes by the magnitude of weights and biases is heuristic. Thus, the contribution seems incremental.

2.	**Lack of theoretical guarantees.** Beyond describing Algorithm 1, the paper provides no formal analysis of convergence, approximation error, or consistency. Questions such as whether the PaTraC posterior converges to the full BNN posterior in the infinite‑width limit remain unanswered. The authors themselves note that a theoretical investigation is future work.

3.	**Small and synthetic experiments.** The main empirical evaluation uses a toy regression with a two‑layer network of 50 neurons and 100 training points. Such small networks may not reflect the behavior of modern deep architectures, where computational savings are most needed. The CIFAR‑10 experiment uses only 500 posterior samples and reports negative log‑likelihoods without accuracy or calibration metrics; the architecture appears to be a simple convolutional or fully connected model, but architectural details are omitted.

4.	**Evaluation metrics and baselines are limited.** Coverage and negative log‑likelihood are informative, but the study lacks metrics on predictive accuracy, calibration, and training cost. In the CIFAR‑10 and Abalone experiments, the deterministic network’s accuracy/error and the full BNN’s performance are not reported beyond NLL. The choice of baselines is narrow. The single line of “Laplace” in Table 1 is unexplained, and no comparisons to simpler scalable methods (MC Dropout, Bayesian last layer, or even Gaussian processes for regression) are given. This makes it difficult to judge the true advantage of PaTraC. Also, a comprehensive ablation study is missing.

5.	**Surprising empirical findings:** In some results, the partial BNNs outperform the full BNN (e.g. lower NLL), which is counterintuitive. This raises concerns that the full BNN might be undertrained or that hyperparameters (prior scales, likelihood variance) are not properly set. The paper does not discuss these anomalies.

6.	**Marginal computational benefit in mixed architecture.** The authors note that the mix‑PaTraC architecture, which uses top‑k Bayesian nodes in every layer, gives little speed‑up in effective sample size per second compared with the full BNN. This undermines claims that partial Bayesianization substantially accelerates inference.

7.	**High computational cost for modest gains.** Even for the small toy regression, 500,000 MCMC iterations were required, thinned to 500 effective samples. For larger networks, the sampling cost is prohibitive. The study does not compare against efficient approximate inference methods (e.g., Laplace approximation, variational inference) that deliver uncertainty estimates at a fraction of the cost.

8.	**Missing discussion of hyperparameter sensitivity and selection.** The choice of the importance measure (squared weights and biases), number of Bayesian nodes k, and prior variances has a significant impact on performance. The paper does not explore the sensitivity of results to these choices nor provide guidelines for practitioners. Additionally, scaling the cross‑entropy loss by 1000 in CIFAR‑10  appears ad‑hoc and is not justified.

9.	**Potential issues with reproducibility.** Critical details such as network architectures for CIFAR‑10, the number of MCMC steps, and random seeds are missing, which hinders reproducibility.

**Questions:**

1.	**Hyperparameter Sensitivity:** How sensitive are the results to hyperparameters such as the prior variances ($\sigma^2_w, \sigma^2_b$) and the ordering exponent α? Did you tune these per experiment, or use fixed values? Including a brief analysis or discussion of hyperparameter choices would be helpful.

2.	**Node Selection Metric:** You select Bayesian nodes based on the sum of squared weights and bias terms (η_i). Have you tried other importance measures (e.g. absolute weight sum, Fisher information)? It would be interesting to know if the results strongly depend on this particular metric, or if other heuristics yield similar performance.

3.	**CIFAR-10 Architecture Details:** The CIFAR-10 experiment reports predictive NLL but lacks details about the neural network architecture (depth, convolutional or fully-connected layers) and training procedure. Please clarify what model was used and why.

4.	**Full BNN vs PaTraC NLL:** In Table 1 (toy regression), the Sep-PaTraC and Mix-PaTraC with 5 nodes have lower NLL than the full BNN. Can you explain why the full BNN performs worse? Could this be due to the choice of burn-in, number of samples, or prior settings?

5.	**Calibration Metrics:** Beyond coverage plots and visual density, have you measured calibration error or width of credible intervals quantitatively? Metrics like expected calibration error (ECE) or average interval width might give a clearer picture of uncertainty quality across methods.

6.	**Multi-Output and Extensions:** How would these methods extend to multi-output tasks or more complex models (e.g. convolutional neural networks)? The selection criterion for output-layer weights assumes equal splitting among outputs; more discussion of multi-output generalization would strengthen applicability.

7.	How does PaTraC compare with modern approximate Bayesian inference (e.g., deep ensembles, Laplace approximations, variational inference)? A comparative study would contextualize the benefits.

8.	**Computational Trade-offs:** The sep-PaTraC BNN achieved much higher ESS/s than the full BNN, but mix-PaTraC did not. Beyond citing code optimization issues, do you have insights into what aspects (e.g. gradient computations, parameter coupling) limit the speed gains? Could approximate inference (like stochastic gradient MCMC) further improve scalability?

Suggestions:

1.	**Provide a deeper theoretical analysis.** Formalize how the PaTraC posterior differs from the full BNN posterior, perhaps by bounding the Wasserstein distance as hinted in the conclusion.

2.	**Expand empirical evaluation.** Evaluate on larger, modern architectures (e.g., ResNet for CIFAR‑10) and include metrics such as accuracy, calibration (e.g., expected calibration error), and wall‑clock training time. Compare against approximate inference baselines.

3.	**Investigate alternative selection criteria and sensitivity.** Perform ablation studies on the importance score, number of Bayesian nodes and prior variance parameters. Provide guidance on selecting k for practitioners.

**Details Of Ethics Concerns:**

I see no ethical issues needing a review.

---

### Official Review · Reviewer_WMvn · 2025-10-30

**Soundness:** 2
**Presentation:** 2
**Contribution:** 2
**Rating:** 2
**Confidence:** 2

**Summary:**

This paper introduces a new family of Bayesian neural network architectures designed to achieve comparable uncertainty quantification to full BNNs at substantially reduced computational cost. The proposed PaTraC BNNs combined the main ideas of trace-class priors with partial BNNs and designed three different network architectures. Empirical studies on toy regression, CIFAR-10, and Abalone datasets demonstrate that PaTraC BNNs retain similar predictive uncertainty to full trace-class BNNs while being faster and achieving higher ESS in some configurations.

[ LLM usage disclosure for this review ]:  I used an LLM to assist in organizing and polishing the wording of my review, but all evaluations, scores, and judgments were entirely my own.

**Strengths:**

1. Combining trace-class priors with partial Bayesianization is innovative and theoretically meaningful.
2. The trace-class prior’s natural ordering offers a principled way to select Bayesian nodes without arbitrary heuristics.

**Weaknesses:**

1. The baseline methods are insufficient. Comparisons are limited only to standard BNNs with a trace-class prior. It would be informative to benchmark against other partial BNNs, variational BNNs (e.g., Bayes by Backprop), ensembles, and SGMCMC methods.
2. The paper does not analyze how close the PaTraC posterior is to the full BNN posterior. A Wasserstein or KL-bound analysis would strengthen the claims.
3. The uncertainty estimation of some PaTraC results is not good enough. For example, as shown in Fig. 4 (Abalone), the Sep variant underestimates uncertainty; the authors acknowledge this but offer no diagnostic insight.
4. No uncertainty calibration or predictive entropy results are provided for the image task. It remains unclear whether PaTraC BNNs can deliver calibrated posteriors on complex datasets.
5. While the ESS/s is improved for some models, the Mix-PaTraC BNN yields almost no speedup compared to full BNNs, which need quantitative justification.
6. Lack of ablation studies, such as Bayesian node selection rules and parameters in the trace-class prior.
7. The writing and structural organization require improvement. There are many grammatical issues and typos, e.g.,  line 169 references Section E in the Supplementary, but no such section exists in the appendix; no description for Table 1 in the main paper, etc. And I believe dedicating a separate section to introduce related work on partial BNNs and trace-class BNNs would be more appropriate.

**Questions:**

1. For CIFAR-10 and Abalone, only NLL is reported. Did the authors analyze posterior calibration metrics (e.g., ECE and predictive entropy) to assess the quality of practical uncertainty?
2. Can the authors provide convergence diagnostics (e.g., R-hat, autocorrelation) or explain why ESS/s sometimes increases with more Bayesian nodes?
3. Could variational or other MCMC methods replace pCNL in PaTraC BNNs while preserving uncertainty quality?
4. Why were strong baselines such as variational Bayes (Bayes-by-Backprop), SG-MCMC, or ensemble methods not included?
5. How sensitive are the results to the Bayesian nodes selection rule? Have the authors tried alternative criteria (e.g., gradient norms, Fisher information, or layer-wise variance)?
6. How sensitive are the results to the $\alpha$ parameter in the trace-class prior and the scaling strategy?
7. Can PaTraC be extended to convolutional or transformer architectures, or is the trace-class assumption inherently tied to fully connected networks?
8. Could the authors derive or empirically estimate a formal measure (e.g., Wasserstein or KL divergence) between the PaTraC and full BNN posteriors?

---

### Official Review · Reviewer_Dt45 · 2025-10-31

**Soundness:** 2
**Presentation:** 1
**Contribution:** 1
**Rating:** 2
**Confidence:** 4

**Summary:**

This paper combines trace-class prior for Bayesian neural network (BNN) and preconditioned Crank-Nicholson (pCN) for fast inference of BNN. The proposal is three NN architectures to decide Bayesian (inference) and non-Bayesian (optimization) nodes. The contribution is incremental. It is not clearly written and contains many grammatical errors.

**Strengths:**

There are some numerical demonstrations on some small benchmark data sets.

**Weaknesses:**

The paper is largely built on trace-class BNN prior (Sell and Singh, 2023) and contains little novelty. The addition of NN architectures to select inference nodes is incremental. There are places confusing or having grammar issues, e.g. 'allows layers to be split into Bayesian and non-Bayesian parameters.' The paper could be strengthened by including comparison with more BNN methods.

**Questions:**

1. How to determine the ordering of NN nodes?

2. How is Out-PaTraC architecture different from last-layer BNN (Snoek et al. 2015)?

3. What is the solid blue curve in Figure 2-(a)?

---

### Note · Authors · 2025-11-21

**Comment:**

We thank all reviewers for their constructive insights and suggestions on how to improve our paper. We believe that the work required to address all the reviewers' concerns is not possible within the review period, and we have thus decided to withdraw our paper to carefully revise our manuscript, in particular the numerical experiments.

**Withdrawal Confirmation:**

I have read and agree with the venue's withdrawal policy on behalf of myself and my co-authors.